# SYSTEM IDENTIFICATION AS A REINFORCEMENT LEARNING PROBLEM

## ABSTRACT

System identification, also known as learning forward models, has a long tradition both in science and engineering in different fields. Particularly, it is a recurring theme in Reinforcement Learning research, where forward models approximate the state transition function of a Markov Decision Process by learning a mapping function from current state and action to the next state. This problem is commonly defined and solved as a Supervised Learning problem. However, several difficulties appear due to inherent complexities of the dynamics to be be learn, for example, delayed effects, high non-linearity, non-stationarity, partial observability and error accumulation when using compounded predictions (i.e., predictions based on past predictions) over large time horizons. In this paper, we elaborate on why and how this problem fits naturally and sound as a Reinforcement Learning problem, and present some experimental results that demonstrate RL is a promising technique to learn forward models.

## 1 INTRODUCTION

One of the most distinguishing features of Reinforcement Learning (RL, Sutton & Barto, 1998) is the direct trial and error interactions between an agent (learner) and its environment (real world, plant, game, etc.), so that the agent can learn the consequences of its actions and thus adapt its behavior to optimize a goal in the long run. However, in real life applications and specifically in industrial settings, there are many mission critical assets where RL cannot be applied in its canonical form, "Online" RL, due to security or operational risks, i.e., the risk of unsafe exploratory actions or the interruption of asset operations. Among recent lines of research in this field, "batch" and "Offline" RL (Lange et al., 2012; Fujimoto et al., 2019; Levine et al., 2020) formulations break down many of these barriers leading to its successful application to these problems.

Our goal is to perform offline RL for these critical real life situations (see dreaming or imagination, Ha & Schmidhuber, 2018; Hafner et al., 2020), however, our method is to apply online RL two times $(RL \circ RL)(data)$: first, for learning good enough forward models, which is the focus of the present paper, and second, for learning a policy using that forward model as the environment. We seek to avoid the well known negative results reported by Fujimoto et al. (2019), while leveraging the exploration ability of RL to learn good policies.

Learning forward models has been an active area of research, with abundant contributions on the application of Machine Learning (ML) (see for instance Werbos, 1989; Fu & Li, 2013; Zhang, 2014; Abdufattokhov & Muhiddinov, 2019; Roehrl et al., 2020). Particularly, it is a recurring topic of research within RL (Sutton, 1991; Sutton & Barto, 1998; Polydoros & Nalpantidis, 2017; Moerland et al., 2020), where forward models usually represent the transition function $s_{t+1} = \mathcal{T}(s_t, a_t)$ of some Markov Decision Process (MDP). We denote an MDP as a tuple $\mathcal{M} = (\mathcal{S}, \mathcal{A}, \mathcal{T}, \mathcal{R})$, where $\mathcal{S}$ is the state space, $\mathcal{A}$ is the action space, $\mathcal{T}$ is the transition function and $\mathcal{R}$ is the reward function, thus, $s_{t+1} = \mathcal{T}(s_t, a_t)$ represents the immediate state after the evolution of the system, starting at time $t$ with state $s_t$ and conditioned by an action $a_t$, and $\mathcal{T}$ is defined by mapping function $s_{t+1} = f(s_t, a_t)$. Learning a forward model is a task commonly defined as a Supervised Learning problem in a direct way (Jordan & Rumelhart, 1992; Moerland et al., 2020), having the set of observations $\mathbf{X} = \{(s_t, a_t), ...\}$, labels $\mathbf{y} = \{s_{t+1}, ...\}$, and a loss, e.g., $\mathcal{L} = ||f(s_t, a_t) - s_{t+1}||$, however, this approach faces several challenges. Here, we propose that this problem has a more complete and natural definition as an RL problem, and show, experimentally, positive results.

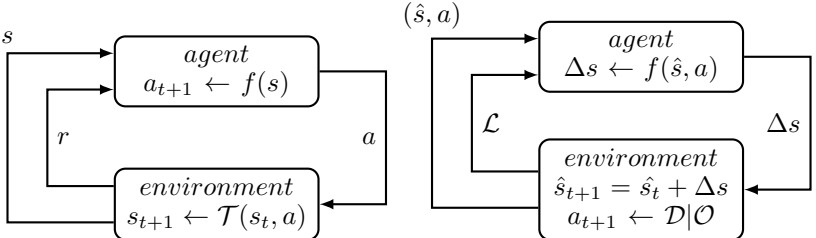

Figure 1: The typical RL setting (left). RL flow for learning a forward model (right).

## 2 MOTIVATION AND PROBLEM DEFINITION

**Why learning forward models with RL?** The domains, tasks, and problems to which forward modeling is being applied are of increasing complexity, including time delayed dynamical effects, high degree of non-linearity, partial observability (POMDPs) and the compounding error issue: a compounded rollout (i.e., generating samples based on past generated samples iteratively) induces a compounding error "when an imperfect model is used to generate sample rollouts, its errors tend to compound - a flawed sample is given as input to the model, which causes more errors, and so on" (Talvitie, 2014). This situation raised the need for additional techniques to adapt the Supervised Learning framework to deal with this increasing complexity, for instance: rollout testing for beyond single step learning robustness, loss accumulation over rollouts for large horizon prediction, recurrent networks or frame-stacking or neural Turing Machines for partial observability, curriculum learning over increasing horizons, data augmentation to aid learning symmetries in data, ensembles of stochastic neural networks to increase the prediction accuracy and reduce bias, and a set of works focused on studying the compounding error (c.f. Oh et al., 2015; Talvitie, 2017; Silver et al., 2017; Fleming, 2018; Asadi et al., 2018; 2019; Xiao et al., 2019; Lambert et al., 2021; 2022).

On the other hand, RL has intrinsic features that provides a natural way to deal with many of those complexities: rollout learning by working with episodic tasks, minimization of the compounding error by optimizing for the long-run by solving Bellman's equation, continuous learning from new experience without requiring a full retraining, and it is by design well suited for stochastic scenarios and works in situations of partial observability.

**Does RL come with an unreasonable extra search cost?** Solving a regression problem by RL has an extra cost due to the required exploration, i.e., searching for a point $y_i \in \mathbf{y}$ (label) that is indeed already known. Thus, why solving this problem with RL? Is it an unreasonable cost? May we have now two problems instead of one?

Let's suppose that we run compounded rollouts of certain length $h$ (time horizon), while minimizing the errors between each predicted point $x_t = f(s_t, a_t)$ and its corresponding true target (label) $y_t \in \mathbf{y}$. Since the predictions in the compounded rollouts trajectories are *sequentially dependent* (by composition), then we face a temporal credit assignment problem, q.e., the root cause of the compounding error. The RL framework deals with this problem naturally through the Bellman's optimality criteria (Bellman 1957). Thus, in this problem setting, we shall not assume that we are given "true targets". Moreover, since we are relying on composition, then, in the Supervised Learning setting, we need the corresponding observation points $x_t$ for the predicted labels $\hat{y}_t = f(x_t)$, however, these are not in the dataset, since $x_t$ are predictions as well. Finally, this extra search cost allows to learn a $Q$-function that learns explicitly the expected prediction error that the model will commit.

**Forward model learning as an RL problem:** Learning forward models with RL can be directly achieved by just translating a regression problem to an RL problem, where the observation (state) is formed by the current state of the system and the previous observed action, the actions of the agent represent the predictions of the next state, and the reward signal is just the negated total/cumulative prediction error. More formally, given an MDP, $\mathcal{M} \coloneqq (\mathcal{S}, \mathcal{A}, \mathcal{T}, \mathcal{R})$, the forward learning problem is defined as the MDP $\mathcal{M}_\mathcal{F} \coloneqq (\mathcal{S}_\mathcal{F}, \mathcal{A}_\mathcal{F}, (\mathcal{D}|\mathcal{O})^\mathcal{F}, \mathcal{L}_\mathcal{F})$, where $\mathcal{D}|\mathcal{O}$ refers to a time series of transitions stored in a dataset ($\mathcal{D}$) or observed ($\mathcal{O}$) from a real world process, $\mathcal{L}_\mathcal{F} = ||s_{t+1} - \hat{s}_{t+1}||$ is

a loss function of the true observed next state $(s_{t+1} \leftarrow \mathcal{D}|\mathcal{O})$ and the predicted next state $(\hat{s}_{t+1})$, $\mathcal{S}_{\mathcal{F}}$ denotes the state space formed by tuples of $(s, a) \in \mathcal{S} \times \mathcal{A}$, and $\mathcal{A}_{\mathcal{F}}$ is the action space defined by elements $\Delta s = \hat{s}_{t+1} - \hat{s}_t$, where $s \in S$. Figure 1 shows the diagrams for the MDPs $\mathcal{M}$ and $\mathcal{M}_{\mathcal{F}}$, respectively. Hence, to apply RL to the forward model learning problem, we just need a "transposition" of the problem and its observations:

$$\mathcal{D}|\mathcal{O} = (s_t, a_t, r, s_{t+1}), \text{ common format of offline RL datasets}$$

$$(\mathcal{D}|\mathcal{O})^{\mathcal{F}} = ((s_t, a_t), \Delta s, r = 0, (s_{t+1}, a_{t+1})),$$

where the final objective is to obtain, by RL, a policy $\pi_{\hat{\mathcal{M}}}$ on the approximated dynamics $\hat{\mathcal{T}} = \pi_{\mathcal{M}_{\mathcal{F}}} \leftarrow \mathcal{M}_{\mathcal{F}}$, such that $\pi_{\hat{\mathcal{M}}}$ is as close as possible to the optimal policy $\pi_{\mathcal{M}}$ in the original MDP, as shown in Equations 1 and 2:

$$\pi_{\hat{\mathcal{M}}} \leftarrow \hat{\mathcal{M}} := (\mathcal{S}, \mathcal{A}, \hat{\mathcal{T}}, \mathcal{R}) \tag{1}$$

$$|\pi_{\mathcal{M}} - \pi_{\hat{\mathcal{M}}}| \propto |\mathcal{T} - \hat{\mathcal{T}}| \tag{2}$$

## 3 METHOD AND DERIVATION

Starting from the one-step definition of a forward model (Eq. 3):

$$\hat{s}_{t+1} = f(s_t, a_t), \tag{3}$$

where $s_t$ is the *state* of the system at time step $t$ and $a_t$ being the control actions applied to the system at $t$, we obtain a one-step (*forward*) prediction of the next *state* $\hat{s}_{t+1}$ of the system. Learning such a forward model $f(s, a)$ implies learning the approximator function $f \approx \mathcal{T}$ with any statistical or ML method (e.g., by fitting the parameters $\theta$ of a parametric estimator $f_\theta$). For this, a loss function is defined as $\mathcal{L}$ (Eq. 4),

$$\mathcal{L} = ||s_{t+1} - \hat{s}_{t+1}||, \tag{4}$$

where $\mathcal{L}$ is minimized during training, for instance, using gradient descent techniques by computing the gradient $\nabla_\theta \mathcal{L}$. Thus, the problem setting naturally appears as a Supervised Learning problem (regression) $\hat{y}_i = f(x_i)$, where the collection of inputs $\mathbf{X}$ is formed by tuples $x_i = (s_t, a_t)$ and the targets collection (labels) $\mathbf{y}$ are commonly defined by the next observed states $y_i = s_{t+1}$. With this definition, a Supervised Learning task is completely defined and can be then easily achieved through a "vanilla" ML pipeline. Now, let the predictions of our model be the deltas $(\Delta \hat{s})$ $w.r.t.$ the last state $(s)$, instead of predicting the new state, thus, Eq. 3 becomes of the following form (Eq. 5):

$$\hat{s}_{t+1} = s_t + f(s_t, a_t), \text{ and hence,} \tag{5}$$

$$\Delta \hat{s} = f(s, a), \tag{6}$$

which describes a well-known recursive relation in the field of Dynamical Systems, by expressing the evolution of the system in terms of its past state plus its derivative (residual).

**Stochastic policies:** Focusing on the most recent works on model-based RL (Janner et al. 2019; Yu et al. 2020), stochastic Gaussian networks are used as the function approximator to learn the policy $f \approx \mathcal{T}$, such that:

$$f(s, a) = \Delta \hat{s} \sim \mathcal{N}\left(\mu_{s,a}, \sigma_{s,a}^2\right) \tag{7}$$

This allows to approximate the dynamics through a stochastic model. There is, perhaps, good margin for improvements over Gaussian networks as pointed out by Chou et al. (2017, i.e., beta distribution), however, the original SAC algorithm, which we use for our experiments, uses a squashed Gaussian Normal Policy for the policy network (see, Haarnoja et al. 2018).

**Compounded rollouts and large horizon loss:** An effective forward model should predict not only the next state, based on the true past state and true action accurately, but should allow as well to simulate the system by running rollouts over its own predictions, implying thus a composition process, that is, predicting the next state based on a previous predicted state (or even a history of predicted states). This condition appears as the most difficult part from the learning perspective, and

it is the cause of the compounding error as well. However, we think that it should be considered as a requirement for a sound and robust definition of a forward model learning problem. Also, some benefits can be obtained which are inherent to a compounded rollout training process, for instance, it creates, trough exploration, an implicit data-augmentation (van Dyk & Meng, 2001; Hernández-García & König, 2018; Iwana & Uchida, 2021) over the source data ($\mathcal{D}|\mathcal{O}$), which is a common practice to improve generalization and reduce over-fitting, but requires an extra work in the supervised ML setting. This implicit data augmentation, instead of enlarging a training dataset from existing data using various translations, acts as a kind of "spatio-temporal" transformation, as in the case of auto-encoders, generating new training data "on-the-fly" (Tu et al., 2018), contributing to obtain robust policies to noisy inputs, aid in learning problem symmetries and generalization, and thus helps to avoid overfitting to the fixed dataset or observations.

Thus, by applying Eq. 8, for running a compounded rollout $E(s_t, (a)_t^{t+h}, h)$ of length $h$ simulation steps, over a fixed sequence of actions $(a)_t^{t+h} = (a_t...a_{t+h})$ and from an initial state $s_t$, we obtain a compounded prediction $\hat{s}_{t+h}$ of length $h$ and a predicted trajectory $\hat{y} = (s_t, \hat{s}_{t+1}, ..., \hat{s}_{t+h})$.

$$\hat{s}_{t+h} = s_t + \sum_{i=0}^{h} f\left(\hat{s}_{t+i}, a_{t+i}\right)\bigg|_{\hat{s}_0 = s_t} \tag{8}$$

Also, equations 5 to 8 define well known recurrent relations that can be seen as the equivalent of recurrent connections, computed with a loop over a sequence input batch where the loss is calculated at the end, as well as dynamical system modeling through Neural ODEs (ODENet, Chen et al. 2018), its augmented version (Teh et al. 2019), and ResNets (He et al. 2016).

Now, a rollout loss $\mathcal{L}_E(s_t, (a)_t^{t+h}, y, \hat{y}, h)$ can then be defined as follows:

$$\mathcal{L}_E(s_t, (a)_t^{t+h}, y, \hat{y}, h) = \sum_{i=1}^{h} ||s_{t+i} - \hat{s}_{t+i}||, \ \ s_{t+i} \in y, \hat{s}_{t+i} \in \hat{y}, \text{ or} \tag{9}$$

$$\mathcal{L}_E(s_t, (a)_t^{t+h}, y, \hat{y}, h) = \mathcal{Z}_E(y, \hat{y}), \tag{10}$$

as a more general definition, being $\mathcal{Z}_E(y, \hat{y})$ a signal (trajectory) similarity function. Thus, in rollout learning, the network is trained with the rollout loss (Eq. 10), instead of the common supervised loss (c.f. Eq. 4). The function $\mathcal{Z}_E(y, \hat{y})$ is an open choice, however, it changes the problem to be solved, as it is the objective function and the goal of the RL problem. In particular, here the problem is sequential and thus the objective is to evaluate how "close" (in terms of shape/pattern recognition) is the predicted trajectory vs. the observed trajectory ($\hat{y}$ vs. $y$). We must note that (r)mse for this problem is just a weak approximation, and that doing signal/pattern recognition using (r)mse alone to measure signal/shape similarity is definitely not a good general measure, see for instance Pandit & Schuller (2019). Also, there are many measures for signal similarity (e.g., KL-divergence, statistical (invariant) moments, signal correlations, etc. ), and usually a combination works better depending on the type of the signals and which signal's features are more important for the problem.

**Policy learning with the Actor-Critic architecture** Finally, a natural way to implement rollout learning is episodic learning, where sequences of transitions are divided in episodes, and rollouts are run over such episodes using loss accumulation for computing the gradient. This approach naturally conducts to thinking on the temporal credit assignment problem (see, Minsky 1961; Sutton 1984; Sutton & Barto 1998), which can be solved effectively through TD-Learning (Sutton 1988). That is, to solve a sequential decision problem optimizing the cumulative sum of a scalar signal over an episodic task. For this, it looks natural to use the Actor-Critic method (Sutton & Barto, 1998; Degris et al., 2012) to train a policy network and, in particular, a stochastic one to predict the deltas (residuals) of the next system state. This can be achieved trough methods like the SAC algorithm (Haarnoja et al., 2018).

## 4 EXPERIMENTAL EVALUATION

We tested the proposed approach on three different MuJoCo environments from OpenAI Gym control suite: Hopper-v2, Walker2d-v2, and Halfcheetah-v2. D4RL (Fu et al., 2020) datasets of the MuJoCo environments have been used to train the forward models. A specific Gym environment

has been developed for training over the trajectories (time series of episodes) contained in the D4RL datasets. The designed Gym environment resets the state and stacked states at each rollout-length step, that is, if we train with a rollout length of $h = 50$ then every 50 steps the state and the whole stack is reinitialized and filled with the ground truth data of the episode contained in the dataset. All MuJoCo environments used here rely on continuous state and action spaces. Further details on the data preparation can be found in Appendix A.4. In MuJoCo, the state variables are divided in two vectors: positions ($qpos$) and velocities ($qvel$). Since positions can be derived from velocities and vice-versa then we opted to predict only the deltas of the positions $\Delta qpos$, and then infer the original velocities of the simulation. The predicted $qpos$ and its corresponding $qvel$ are calculated according to:

$$qpos_{t+1} = qpos_t + \Delta qpos \tag{11}$$

$$qvel_{t+1} = \frac{\Delta qpos}{dt} \tag{12}$$

The observations (inputs) for the models are windows (stacks) of states of length $w = 20$ (past+current), that is, the well-known frame-stacking method. The observations contains as well the respective actions taken at each time step as stored in the episodes of the D4RL datasets. Also, for adding more variability to the training trajectories data, each episode starts randomly from a time step in the range $[0, 30]$, so that the model will not see the same sequences all the time but random shifts of the episodes.

We use the Soft Actor-Critic (SAC) algorithm (Haarnoja et al., 2018) as the RL algorithm, thus the actor networks policies represent the learned models. For all the experiments, the same set of hyperparameters for the Actor and Critic are used: encoder architecture is an MLP of 6 hidden layers of 512 elements each with Mish (Misra, 2019) as activation function. Quantile regression (Dabney et al., 2017) is used for the $Q$-function networks (with $n$-*quantiles* $= 64$). For each network, and learning step, the batch-size used is 1024. Networks inputs are min-max scaled based on the datasets. Other information about specific training parameters like the number of episodes run per experiment, or gradient steps can be found at Table 1. Refer to Appendix A.2 for additional details about hardware and software used.

Table 1: Experiment's training parameters and main test metrics results.

| Environment | Training Parameters | | | Testing - last values (average) | | |
|---|---|---|---|---|---|---|
| | Episodes | #Params | Grad. Steps | Cr. loss | Act. loss | rmse |
| Hopper-v2 | $12K$ | $1425K$ | $\sim 121K$ | 0.49 | 2.05 | 0.09 |
| Walker2d-v2 | $12K$ | $1488K$ | $\sim 122K$ | 5.76 | 10.63 | 0.19 |
| Halfcheetah-v2 | $12K$ | $1488K$ | $\sim 121K$ | 69.06 | 33.00 | 0.28 |

For these experiments, we used a pseudo-sparse reward signal as follows:

$$r_t(s, a) = \begin{cases} 1 - \mathcal{Z}_E(y, \hat{y})(\text{signal similarity function}) & \text{at rollout ends,} \\ -\|s_t - \hat{s}_t\|_{L_2} & \text{otherwise,} \end{cases} \tag{13}$$

being the signal similarity function $\mathcal{Z}_E(y, \hat{y})$:

$$\mathcal{Z}_E(y, \hat{y}) = \left(1 + \|y - \hat{y}\|_{L_2}\right)\left(1 + \|\nabla y - \nabla \hat{y}\|_{corr}\right)\left(1 + \|y - \hat{y}\|_{KL}\right), \tag{14}$$

where, $\|y - \hat{y}\|_{L_2}$ is the sum of the squared error between true and the predicted values, $\|\nabla y - \nabla \hat{y}\|_{corr}$ is the correlation distance between the derivatives of both predicted and true values and $\|y - \hat{y}\|_{KL}$ is the Kullback-Leibler divergence between both true and predicted values, over all the the current rollout steps.

For Hopper, Figure 2 shows the overall RMSE of predicted variables for the forward models. After running a total of 12K episodes and $\sim$121k gradient steps, the overall RMSE converged close to 0.10. Additionally, both actor and critic loss curves show that the SAC agent is learning a good policy from episodes 1000 to 4000 (see 2 for additional experimental results). After these episodes, both curves converge asymptotically, suggesting a pseudo optimal policy has been achieved. Walker2d has similar number of gradient steps and episodes as Hopper, it is noticeable how errors are higher

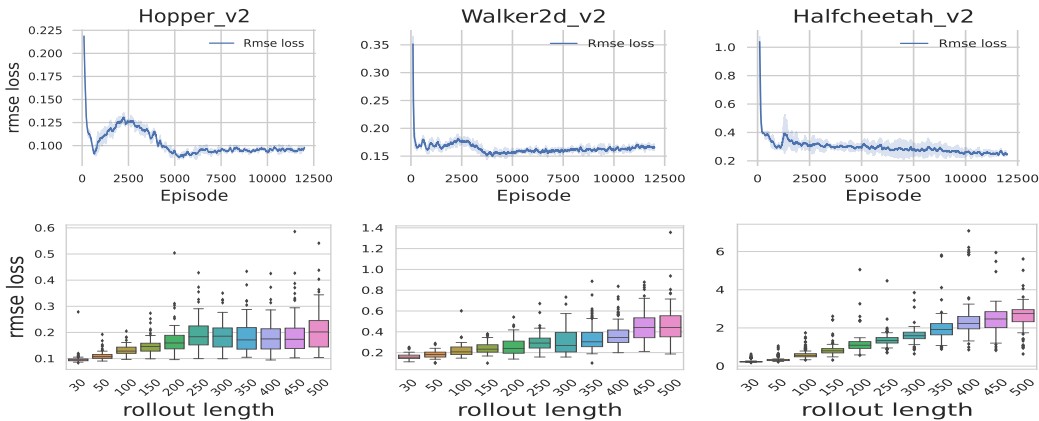

Figure 2: Hopper, Walker2D and HalfCheetah: RSME per episode and rollout test performance

in terms of RMSE (Fig. 2) as well as actor and critic losses (see Appendix 2) than Hopper. This is basically due to the higher number of predicted variables (almost double), and the higher complexity of the environment. We can also notice an error increase when the model is tested with higher rollout-steps. However, the asymptotic behavior of these curves is still maintained, suggesting the finding of a stable sub-optimal policy. Finally, HalfCheetah is the hardest environment tested. Despite having the same number of target variables than Walker2d, these are noisier as the dynamics of the robot is more complex and the results seems worse than in the other two.

### 4.1 COMPARISON OF REINFORCEMENT LEARNING VS. "VANILLA" SUPERVISED LEARNING

To compare both Supervised Learning and Reinforcement Learning approaches we selected the Hopper environment and designed a specific setup.

**Reinforcement Learning setup:** Two SAC based RL forward models have been trained (RL-v1 and RL-v2) with different stack windows. RL-v1 uses a stack of $w = 10$, while RL-v2 uses $w = 30$. The reward function used for this experiments is a fully-sparse reward, that is, $r = 0$ for every time step, and $r = $ signal similarity measure (between both true and predicted trajectories) at every rollout end. This has been designed in this way to test the RL approach in it's extreme version, despite this sparse reward function will hurt convergence speed. The signal similarity measure (RL goal) used in this experiments is a simplified but effective similarity measure. It incorporates explicit measures from time-domain, frequency-domain and power-domain measures (c.f. Ivan & Brian, 2022). The rollout length for training was $h = 50$ episode steps.

**Supervised Learning setup:** Two Supervised Learning models (SL-v1 and SL-v2) have been trained and tested with different stack windows as well. SL-v1 has a stack of $w = 10$, while SL-v2 uses $w = 30$, same as RL setup. The Supervised Learning experiment follows the same structure of steps/rounds of the RL training setup. In particular, we use the same SAC Actor Networks for the Supervised Learning policy, however, samples are taken from the ground truth DRL4 datasets instead of a replay buffer as RL does. Each sample is of $batch - size = 1024$ and, at every step/round, 100 mini-batches are randomly sampled to train the network using the MSE Loss.

Thus, both approaches use the same Policy Networks, but trained using two different methods. Special attention have to be paid to the fact that these two approaches solve different problems, since they have different optimization objectives: the RL training uses a signal similarity function over entire rollouts as its goal, while the SL training optimizes the MSE over random mini-batches.

**Experimental results:** A total of 25.000 steps/rounds have been performed per experiment. Figure 3 shows four different plots: Critic Loss (upper-left) of the RL approach per evaluation-round, Critic Loss for SL is 0 since no critic is trained for the SL experiments. Supervised loss MSE

(upper-right) per evaluation-round for all policies (RL+SL) since it is possible to evaluate an RL-actor policy network over the ground truth DRL4 datasets in the same way as the SL policy. RMSE rollout-metric (lower-left) which is the RMSE evaluated at the current evaluation-round over all the steps of a random episode $e_r$ from the dataset, comparing the predicted trajectory of the entire episode vs. the true trajectory of the entire episode. Mean rollout reward (lower-right) shows the mean of the rewards obtained by the evaluation of the signal similarity measure at each rollout end of the random episode.

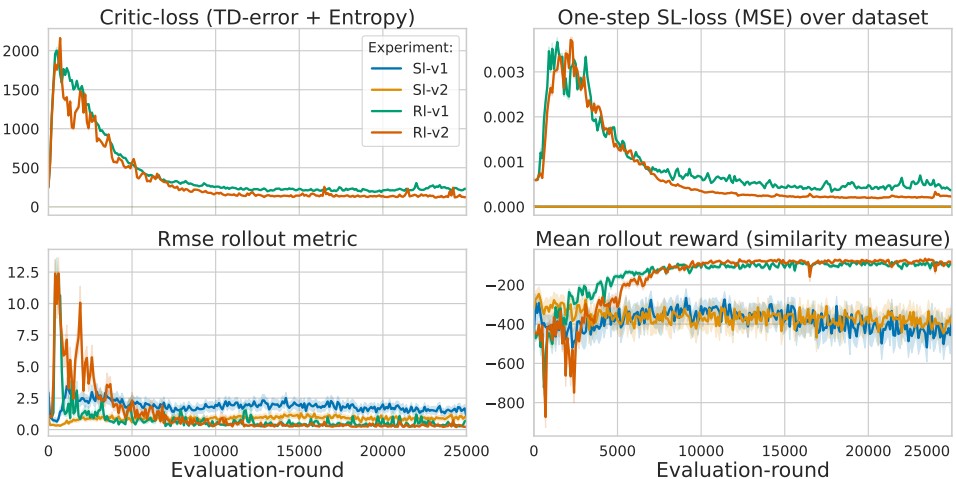

Figure 3: Supervised Learning vs. Reinforcement Learning training and evaluation metrics.

Some conclusions can be elaborated from observing the results in Figure 3: SL trained policies converge very fast to very low values of the Supervised loss MSE and low values of the RMSE rollout-metric (SL-v1 and SL-v2 lines almost converge in the firsts training rounds), however, it is not the same for the mean rollout reward (signal similarity measure) and the SL-v2 version ($w = 30$) achieves better optimality than SL-v1 ($w = 10$). The RL trained policies convergence speed is notably slower (orders of magnitude), however both versions of RL policies achieves better optimality than SL in RMSE rollout-metric and Mean rollout reward. Although RL-v2 policy of ($w = 30$) shows a better optimality in the RMSE rollout-metric than the RL-v1 ($w = 20$), it is not clear that this is true for mean rollout reward metric. It is somehow natural that the trained SL policies does not perform well for the mean rollout reward metric, since they are not trained with this learning objective, however, both RL policies are also converging to low enough values of the Supervised loss MSE metric, even when they are not trained with this objective.

Finally, from the whole figure, it can be observed that beyond some "rounds" RL starts to gain advantage over the vanilla SL approach, not only achieving better values in the RMSE rollout-metric and mean rollout reward metrics, but also with significantly less standard deviations, as shown in the shaded regions enclosing all curves.

Also, with the corresponding trained policies, we tested each one by predicting trajectories on random episodes for different rollout lengths: $h \in \{50, 100, 200, 300, \ldots, 1000\}$.

By observing the results in Figure 4: It is noticeable how the SL policies (v1 and v2) perform better than RL on shorter rollouts ($h \lesssim 300$), however, as we increase the length of the rollouts, the SL trained policies suffer an exponentially increasing compounding error. Nevertheless, the RL approach seems to be more robust against rollout lengths, suggesting, perhaps, that RL polices may be able to better resist strong exploration and planning over them without diverging.

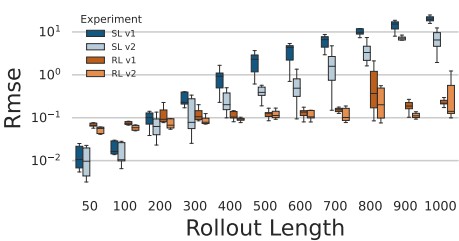

Figure 4: Increasing rollout size testing for the learned policies (log scale).

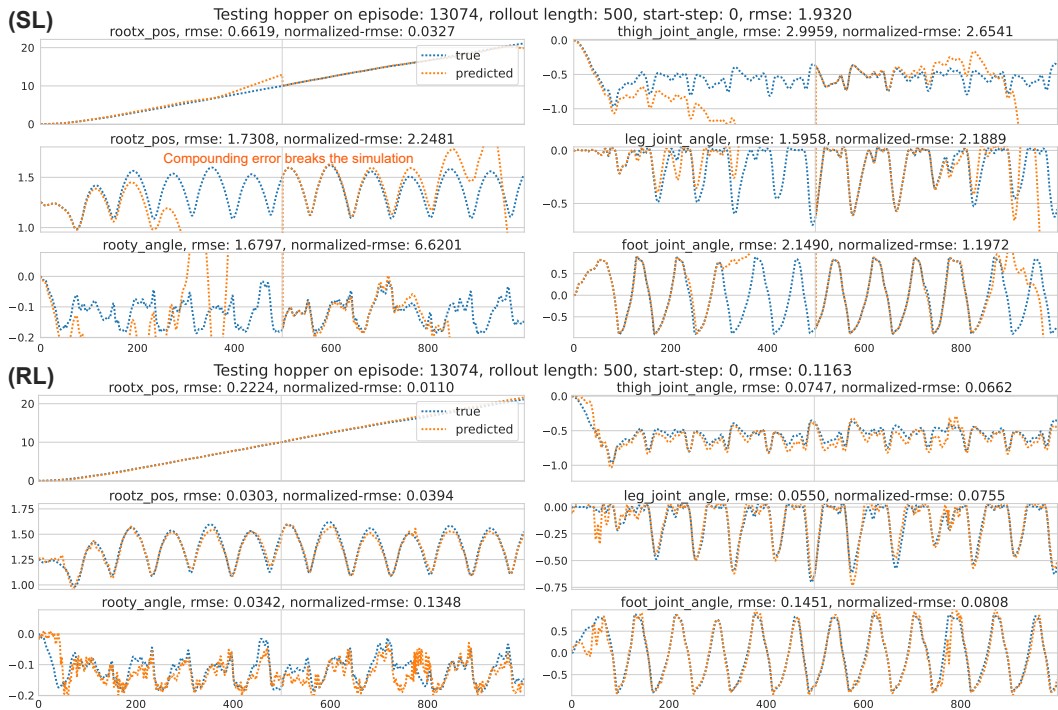

Figure 5: Above: Random episode of rollout length 500 steps for the Supervised Learning policy. Below: Random episode of rollout length 500 steps for the Reinforcement Learning policy.

Finally, Figure 5 shows a random episode comparing the true vs. predicted trajectories using rollouts of $h = 500$ steps, for both SL-v2 and RL-v2 policies. It can be seen how the compounding error increases for the SL policy up to the point that it completely becomes useless. On the contrary, the RL policy manages to control the compounding error and keep predictions in range. More importantly, in the RL policy now we can't talk about a systematic compounding error, since it is possible for the RL policy to have an error $E > e$ in some steps and have $E < e$ in subsequent steps.

## 4.2 INITIAL RESULTS ON TRAINING RL AGENTS ON FORWARD MODELS

To investigate the behavior of the learned forward models (SL, RL) as training environments, a standard RL training procedure using Gym was performed. These experiments were run using the well known tianshou (Weng et al., 2022) library examples. The mujoco Hopper-2 SAC algorithm example (examples/mujoco/mujoco_sac.py) was used with an additional parameter to specify a different environment for testing the agent, in a way that the evaluator performs the tests on the real Gym Hopper-v2 environment instead of the forward model environment.

From the results of testing the SL-v2 forward model (Figure 5), we have seen that it is clearly affected by the compounding error, and also, by observing Figure 6, we see how this affects the training process producing very short episodes and small training rewards. A hypothesis to explain this is that the SL-v2 model protects itself from data it has not seen in its training dataset, cutting the episodes earlier due to an exploding compounding error (Hopper-v2 explicitly cut episodes when variables are out of band), and thus admitting only inputs that are very close to the data in its training dataset, hence, data close to the real environment. However, for environments that do not penalize out-of-band variables this can be a strong problem. Based on the previous hypothesis, the forward model RL-v2, as observed in Figure 5, does not suffer a severe compounding error, and thus, it should be able to admit more diverse inputs and longer episodes without an exploding compounding error, as it effectively occurs in the experiments (Figure 6). However, this robustness to "bad" inputs may come with the possible drawback of not cutting episodes which are clearly artificial and impossible to exist in the real environment, affecting severely the learning process. This is a known

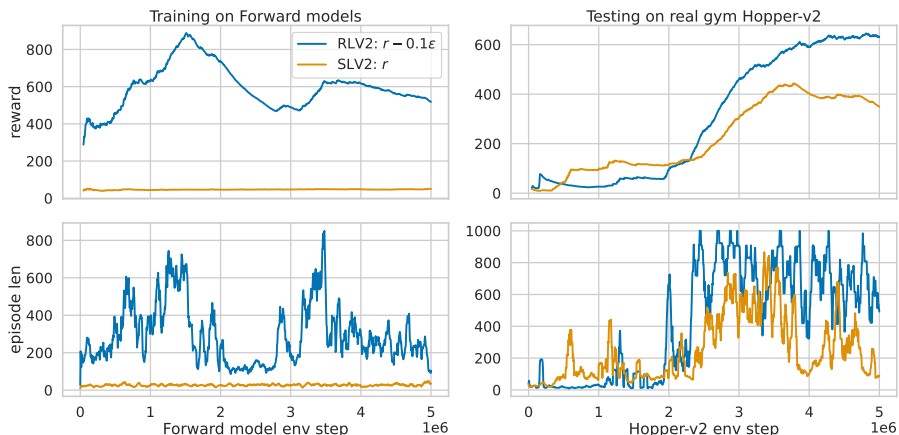

Figure 6: Results of training an SAC agent in an environment whose dynamics are based on the forward models SL-v2 and RL-v2. At each epoch, test runs are performed over the "real" gym Hopper-v2 environment.

common concern: an agent trained over a learned forward model may exploit weaknesses of the model to gain unreal advantages in optimizing the reward function (the cheating effect).

Hence for the SAC agent trained on the RL-v2 forward model, we opted for penalizing the reward ($r$) with a model uncertainty measure. In particular, the reward is penalized by $r - 0.1\epsilon$, where $\epsilon = Q_f((s, a), \Delta s)$ is the expected error of the model prediction. This penalization is an attempt to aid the learning agent to focus on optimizing transitions for which the forward model has a low error, and thus, are closer to the real environment. This idea has been applied as shown also in Figure 6.

## 5 CONCLUSION

This paper presents a framework to learn forward models with Reinforcement Learning. We have shown significant reasons on the benefits to consider forward-model learning as an RL problem. We have defined and described how to model this problem with RL, and we have tested the proposed method over three well-known environments from the MuJoCo Gym collection, compared it against a commonly-used Supervised Learning setup, showing significant results in reducing the compounding error for large horizon simulations. Finally, we have shown the results of training RL agents over these RL-based forward models with promising results as well. We think that the main possible reasons for these positive results can be that: 1. RL controls the compounding error by optimizing for the long-run solving a Bellman's equation; 2. The exploration of RL during training on compounded rollouts enriches the robustness of the policy to noise and diverse inputs, and thus, it reduces the compounding error as well; 3. Using a similarity function on whole rollout trajectories is a preferred objective over the widely adopted one-step $\ell_p$-norms as a loss function.

The guiding objective is the use of these forward models to train RL control agents to leverage exploration in offline RL, with the aim to obtain results which are more comparable against agents trained in "real" environments. Also, we are doing research on how to exploit the $Q$-function learning by RL during the policy training. This $Q$-function learns explicitly to estimate the expected cumulative error the model will commit in its predictions, an information which is not commonly present in many of the current available methods. That information gives an explicitly learnt uncertainty measure on the model's predictions, and thus it can be very helpful when exploited by model predictive control methods or Reinforcement learning algorithms, e.g., by weighting rewards or any other exploratory or penalizing scheme. We presented as well initial experimental results on this line with promising results.

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

## A    APPENDIX

### A.1    EXTENDED DISCUSSION AND CONCLUDING REMARKS

**Real Life motivation**    Reinforcement Learning is a very powerful technique for solving Industrial Control Problems. However, in the Industrial setting, there are many mission critical assets where RL cannot be applied in its canonical form, for security or operational risks, i.e., the risk of unsafe exploratory actions or the interruption of asset operation. Among the last lines of research on this matter, the "Offline Reinforcement Learning" formulation drops many barriers in the successful application of RL to these problems. Offline RL uses observations from the true industrial asset operation trough a dataset of trajectories (including control actions). This dataset is then used to perform offline RL to learn a control policy.

However, it is very common, by causes of operational reasons, that not all control actions are well represented in the dataset (out of distribution "OOD" actions), and thus the estimated value of such actions may be largely over/under estimated. The reason is that the returns of such OOD actions can not be learnt (penalized/rewarded) during training, since there is no interaction with the true environment. Hence, OOD actions become untested actions with maximal uncertainty in its estimated values. This fact, induces poor (or completely wrong) policies withing the canonical RL setting, and the performance of the current state of the art Offline RL algorithms is being actively investigated. However, if a forward model is provided then RL algorithms can explore on it, and avoid or mitigate the problem of ODD actions.

Along this line, the natural improvement over the presented results, is being able to understand how good or bad our forward models represent the real environment dynamics (beyond the information contained in source datasets and error metrics). In order to analyse this key point, the goal is to train standard RL agents on these models and compare their performance against real environment trained ones.

Although we have done initial steps on this direction, setting up, and controlling the experimental setup of such experiments must be further studied, since our initial tests did not yield the expected results. We argue that once these tests are completed, their results will give us a better understanding of the robustness of our approach.

**The claim**    First, we want to remark that the presented comparison between Supervised Learning and Reinforcement Learning is not about algorithms, but problem paradigms. We do not deny that Supervised Learning, with enough extra features, can be used to solve the problem as effectively as any Reinforcement Learning approach or even better. However, what we argue is that the procedure to build such specialized Supervised Learning method signals the quest to solve a problem that is more naturally stated as a Reinforcement Learning problem or a general sequential decision problem.

**The proposed method**    There are many other aspects to consider for further optimization of the learning procedure, for instance:

**Does this problem impose a special constraint to the $Q$-function?**    Since we know that a sequence of perfect actions should return 0 as rollout error, it can be argued that we know in advance the right returns of many $(s, a)$ pairs if we do not consider signal noise and a source of randomness in the collected data. We can even consider that these sources exist, but are negligible so we can then explicitly impose such constraints into the learning of the $Q$-function.

Along this line, one tested trick is to select random tuples $(s, a, s', r = 0)$ from the datasets and update the critic network with such tuples with the aim of improving learning convergence. However, in our initial experiments, using this trick cause the $Q$-function to diverge if the ratio of this updates vs. the updates from the replay buffer is not controlled.

**Transferring the knowledge from the policy to the $Q$-function**    Another trick is to try training the policy network via Supervised Learning with some random $(sa, s')$ pairs so that the policy can take advantage of Supervised Learning as well. But this will require a special procedure to update the critic network specifying confidence on some $(s, a, R)$ tuples. This looks analogous

to the procedures used by offline-RL algorithms to constraint the $Q$-values on out-of-distribution (OOD) points. We have tested this trick by helping the policy learning using random $(sa, s')$ to try to improve learning, however without a clear procedure to update the critic network as well, there is a fierce competition between the updates to the policy to optimize the critic and the Supervised Learning ones. Besides, these results show also a divergence in the $Q$-function and the policy learning.

**Avoiding the hacker agent against a learned model** A known common concern is the idea that an agent trained over a learned forward model may exploit weaknesses of the model to gain unfair advantage in optimizing the reward function (the cheating effect). Some authors have argued that this can be alleviated due to the inherent noise contained in the learned forward-model. Another simple idea is to over constraint the reward function for the learned forward models. We think that a future research direction is how to prevent this cheating effect. However, in our early experiments we can not conclude that the agents systematically gain unfair advantages. Instead, what we observe is just slightly different behaviors that of course yield different returns, but not a clear trend to optimize over the model, degrading the performance in the original environment.

## A.2 Hardware and Software

For all the experiments we have used a Linux virtual machine with enough system RAM and 4 Nvidia Tesla-T4 GPUs. The experiments presented here rely on the following Deep Reinforcement Learning specific software: for the environments (problems to solve) OpenAI Gym (Brockman et al. (2016)) and MuJoCo (Todorov et al. (2012)) are used. For Deep RL algorithms, the SAC implementation in the d3rlpy library (Takuma Seno (2021)) is used. For implementing a replay buffer we do not use the d3rlpy's replay buffer implementation, instead the cpprb library (Yamada (2019)) is used. Finally, as the Deep Learning backend and framework, Pytorch (Paszke et al. (2019)) is used. D4RL datasets (Fu et al. (2020)) are used as as the source of dynamics trajectories. Every library and asset mentioned in this paper and used in this research has the license to be used without restriction.

## A.3 Environments' spaces

In this section, we present the list of variables of the state spaces for both the original problem $\mathcal{M}$ and the forward model approach $\mathcal{M}_{\mathcal{F}}$ for the three MuJoCo environment used in the experimental evaluation. We recall that in the case of the forward model environment we have decided to predict only the position's state variables (target variables) because the velocities can be derived from them in order to reduce complexity of the problem. Those variables have been marked with an (*).

Table 2: State space variables for MuJoCo and derived forward environments

| Hopper | Walker2d | Halfcheetah |
|---|---|---|
| rootx pos* | rootx pos* | rootx pos* |
| rootz pos* | rootz torso* | rootz pos* |
| rooty angle* | rooty torso angle* | rooty pos* |
| thigh joint angle* | thigh joint angle* | bthigh angle* |
| leg joint angle* | leg joint angle* | bshin angle* |
| foot joint angle* | foot joint angle* | bfoot angle* |
| | thigh left joint angle* | fthigh angle* |
| | leg left joint angle* | fshin angle* |
| | foot left joint angle* | ffoot angle* |
| rootx vel | rootx vel | rootx vel |
| rootz vel | rootz vel | rootz vel |
| rooty angle vel | rooty angle vel | rooty angle vel |
| thigh joint angle vel | thigh joint angle vel | bthigh angle vel |
| leg joint angle vel | leg joint angle vel | bshin angle vel |
| foot joint angle vel | foot joint angle vel | bfoot angle vel |
| | thigh left joint angle vel | fthigh angle vel |
| | leg left joint angle vel | fshin angle vel |
| | foot left joint angle vel | ffoot angle vel |

## A.4 TRAINING DATA PREPARATION

Training data for the forward model have been extracted from D4RL repositories (Fu et al. (2020)). For each example, several datasets from this library have been included in order to get a large and diverse collection of trajectories (see Table 3). Additionally, and for training purposes they have been filtered only those experiences with episodes longer than a minimum number of steps (500 for the main experimental results and 100 for the additional tests). This filtering aims to find longer and more stable episodes to learn.

Table 3: D4RL source datasets used per experiment.

| Hopper-v2 | Walker2d-v2 | Halfcheetah-v2 |
|---|---|---|
| hopper-random-v2 | walker2d-random-v2 | halfcheetah-random-v2 |
| hopper-medium-v2 | walker2d-medium-v2 | halfcheetah-medium-v2 |
| hopper-expert-v2 | walker2d-expert-v2 | halfcheetah-expert-v2 |
| hopper-medium-replay-v2 | walker2d-medium-replay-v2 | halfcheetah-medium-replay-v2 |
| hopper-medium-expert-v2 | walker2d-medium-expert-v2 | halfcheetah-medium-expert-v2 |

## A.5 TRAINING PROCEDURE

Here, you can find the main algorithms of the presented forward models (Algorithms 1, 2, and 3).

---
**Algorithm 1** Forward model main training loop
---
1: Initialize replay-buffer ($RB$), SAC algorithm ($SAC$) and forward model GYM environment ($FW$)
2:
3: total steps = 0
4: **for** $episode$ **do**
5:     steps = 0
6:     total reward = 0
7:     results = collect($FW$, $SAC$) {(described at algo.2)}
8:     **for** sample in results **do**
9:         $RB \leftarrow$ sample {(append)}
10:         steps $+ =$ sample[steps]
11:         total reward $+ =$ sample[reward]
12:     **end for**
13:     total steps $+ =$ steps
14:     **for** $i$ in range(10) **do**
15:         samples = $RB$.sample($SAC$.batchsize)
16:         loss = $SAC$.update(samples)
17:     **end for**
18:     Update metrics
19:     **if** $episode$ % 100 == 0 **then**
20:         Save SAC model and policy
21:     **end if**
22: **end for**
---

---

**Algorithm 2** Collect Algorithm

---

1: Initialize $buffer$. env and SAC included as argument.
2:
3: state = env.reset()
4: **for** $n$ in $range(10000)$ **do**
5:    **if** explore **then**
6:       action = SAC.sample(state)
7:    **else**
8:       action = SAC.predict(state)
9:    **end if**
10:    step = env.step(a) {described at algo.3}
11:    state = step[s]
12:    $buffer \leftarrow step$ {append}
13: **end for**
14: **return** $buffer$

---

**Algorithm 3** Forward model environment step algorithm

---

1: Initialize observation as $stack$, predicted state as $buffer$ and true state as $buffer$. dataset included as argument, with the real experiences from d4rl datasets. action performed by SAC agent (sac action) is passed as argument.
2:
3: step counter $+ = 1$
4: rollout step counter $+ = 1$
5: true obs = dataset[step counter][observation]
6: true action = dataset[step counter][action]
7: predicted state = observation $+$ sac action
8: observation $=$ predicted state
9: **if** rollout step counter $>=$ rollout step **then**
10:    observation = true obs
11:    rollout step counter $= 0$
12:    rollout terminal = True
13: **end if**
14: predicted $\leftarrow$ predicted state, action, rollout terminal {append}
15: true state $\leftarrow$ true obs, true action, terminal {append}
16: **if** step counter $>=$ total episode length **then**
17:    terminal = True
18: **else**
19:    terminal = False
20: **end if**
21: reward = getReward(predicted state, true state, terminal)
22: **return** observation, reward, terminal

---

### A.6 ADDITIONAL EXPERIMENTAL RESULTS

In this section, there are some detailed additional results that may result useful in order to understand not only the training stage (i.e., actor and critic losses figures, as well as detailed training main metrics Table 1), but also the testing of our forward model on dataset actions.

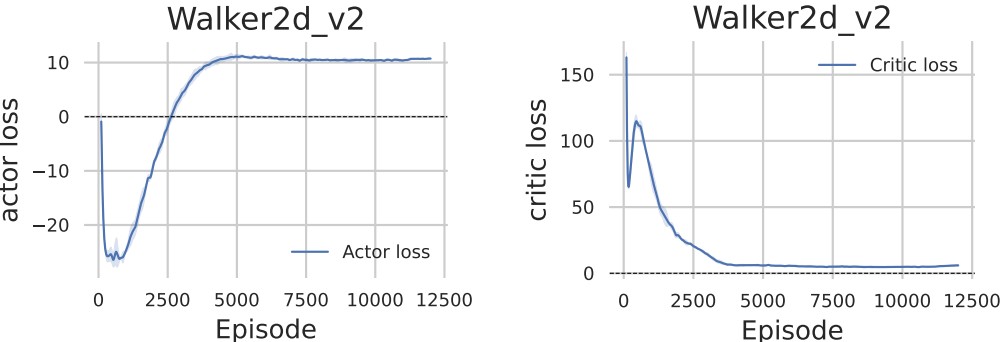

Figure 7: Walker2d SAC Actor loss and Critic loss (training) per episode

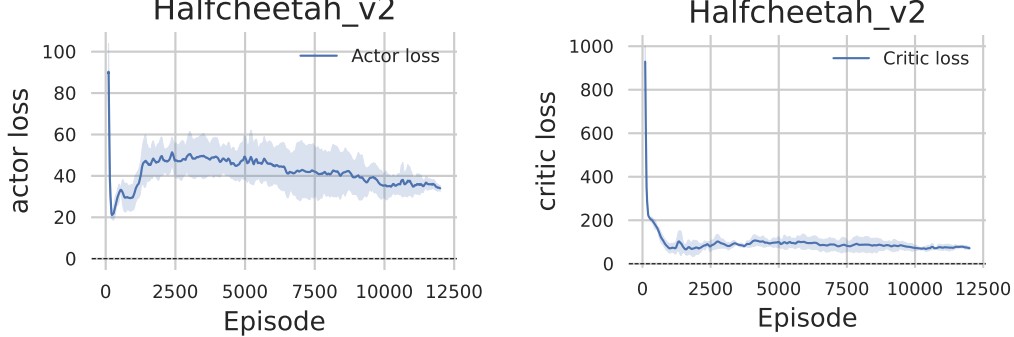

Figure 8: HalfCheetah SAC Actor loss and Critic loss (training) per episode

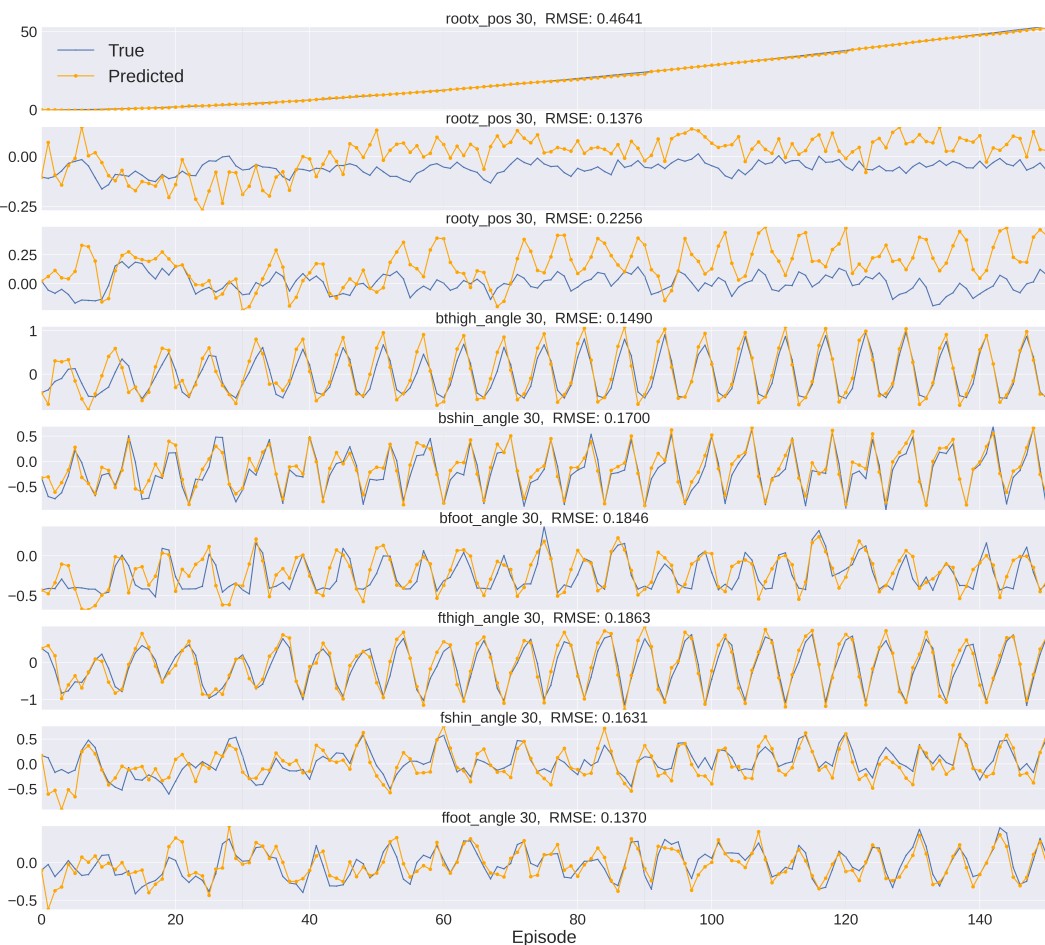

Figure 9: Halfcheetah forward model prediction of target variables, compared to real values. Roll-outs of 30 steps. Sample of 150 steps (5 full rollouts of 30 steps each) obtained from a full random episode.

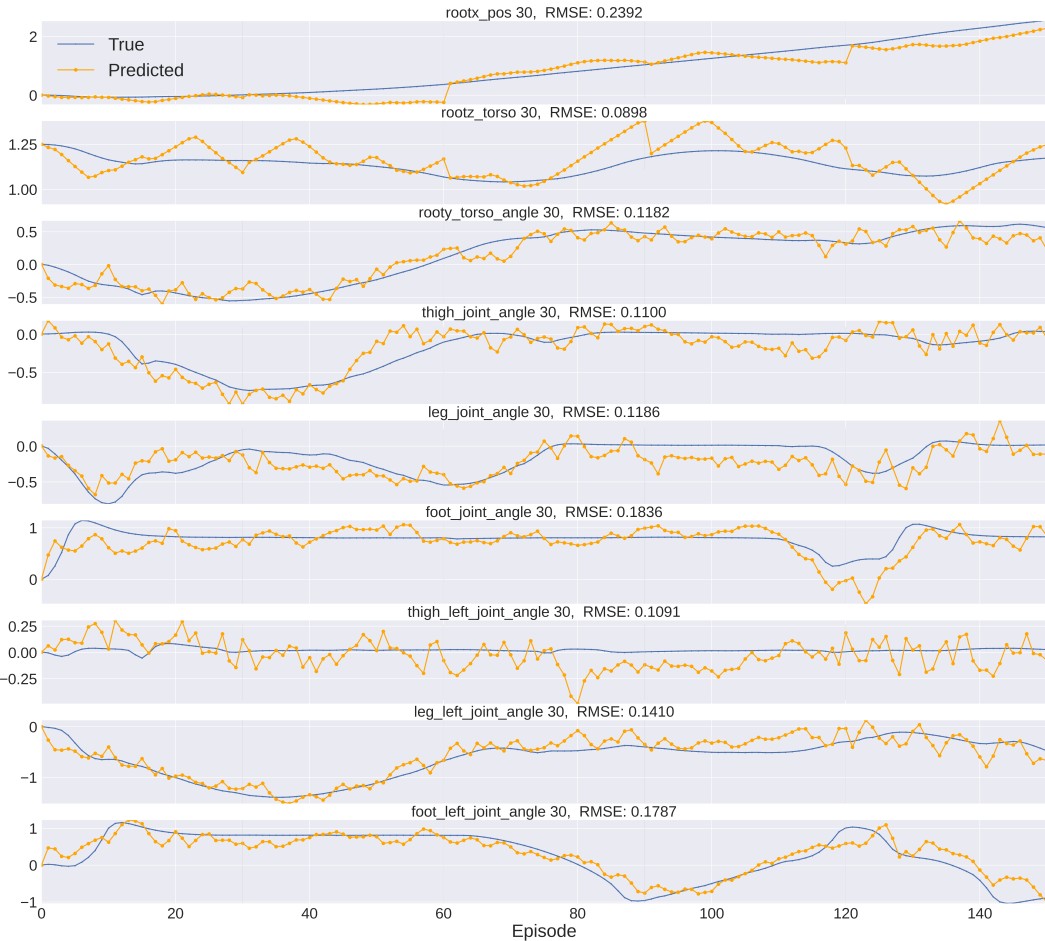

Figure 10: Walker3d forward model prediction of target variables, compared to real values. Rollouts of 30 steps. Sample of 150 steps (5 full rollouts of 30 steps each) obtained from a full random episode.

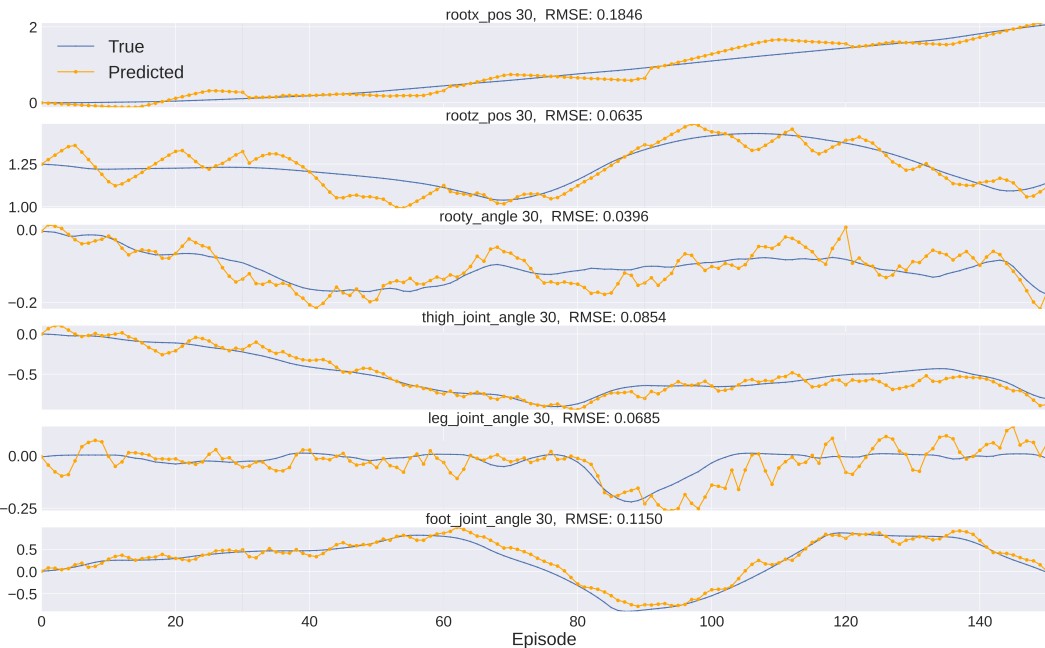

Figure 11: Hopper forward model prediction of target variables, compared to real values. Rollouts of 30 steps. Sample of 150 steps (5 full rollouts of 30 steps each) obtained from a full random episode.

