# OpenReview forum: "System Identification as a Reinforcement Learning Problem"
_ICLR.cc/2023/Conference — Submitted to ICLR 2023_

### Official Review · Reviewer_NoVW · 2022-10-23

**Confidence:** 4
**Correctness:** 3
**Technical Novelty And Significance:** 3
**Empirical Novelty And Significance:** 3
**Recommendation:** 5

**Clarity, Quality, Novelty And Reproducibility:**

**Clarity**

The method sections of the paper and descriptions of the experiments are generally well-written. The experiment sections starting at Section 4.1 has some weaknesses related to clarity of the meaning of the results.

**Quality**

In its current form, it is difficult to determine the quality and significance of the results and potential contributions of the method.

**Novelty**

The novelty of the calculating forward models with RL appears sound, but is limited without showing how those kind of models would perform in applications.

**Reproducibility**

The paper appendix provides detail on experiment setups, but the paper does not include a reproducibility statement or code in the supplementary material.

**Strength And Weaknesses:**

**Strengths**

* The paper provides a novel formulation to train forward models in RL settings. The authors outline their method, its formulation, and relevant parts in great detail for the reader to follow.

**Weaknesses**

* The experimental data and the authors description of it does not provide a set of clear conclusions about the method. It would be good if the authors could provide greater clarity into the main conclusions from their experiments.
* It's unclear how the authors proposed method would perform (RL calculated forward models) would perform in relevant application settings, such as model-based RL. The authors allude to this point in the conclusion and future work, but the paper would be greatly strengthened if indications of performance would be included.
* Following on the above two points, the significance of the overall results is unclear. It would be good to add further clarity to this, ideally by additional experiments in an application setting or an indication of the advantages this method could provide.

**Additional Questions**

* What is dt in Table 1? I do not see it defined in the text near it.
* Where is the reward in Figure 2? Is it the same as the RMSE, which would be confusing since it should be the signal loss, correct?
* Could you clarify test rollout setting - is the model just predicting further into the future?
* It's unclear what Figure 4 and Figure 5 are trying to show. Could you provide further description?

**Summary Of The Paper:**

The paper introduces a reinforcement learning (RL) based formulation for learning forward models, which is often referred as learning the dynamics of an environment in a control setting. The authors first motivate the desire for computing forward models and then highlight some of the issues with current approaches to forward model computations that heavily on supervised learning setups. The authors the describe some of the potential advantages of computing forward models with RL and introduce a formulation to achieve that. Next, the authors introduce relevant definitions and design choices for their method, such as the choice of the policy (SAC-based Gaussian policy), the rollout loss based on trajectory similarity functions and a motivation for using Actor-Critic methods by leveraging ACs ability to resolve temporal credit assignment.

In their experiment, the authors study three different D4RL environments (Hopper-v2, Walker2d-v2, Halfcheetah-v2) and their specific forward model definition based on calculation of deltas of positions between different states. The first set of experiments outlines how well a RL trained model performs (measured in rmse loss) on three environments. The authors then compare their RL-based formulation to a supervised learning formulation for the same three environment and study the performance of each method. The results show that RL method generally takes longer to train and does not perform as well in RMSE as the supervised learning. The RL method does appear to have a lower error with longer rollout length compared to supervised learning.

**Summary Of The Review:**

In its current form, the reasons to reject outweigh the reasons to accept the paper. I think that the paper could be significantly strengthened by providing greater clarity as to the significance of the results and its broader implications.

---

> ### Author Response · Authors · 2022-11-18
> **Thank you, we have prepared a new version taking into account all the very helpful comments of all reviewers**
>
> Thanks in advance for your time, we appreciate your comments. Please, find below the answers to your questions and comments.
> Weaknesses:
>
> >The experimental data and the authors description of it does not provide a set of clear conclusions about the method. It would be good if the authors could provide greater clarity into the main conclusions from their experiments.
>
> * Answer: Concerning the conclusions, we are going to upload a new version of the paper with both things extended. We have included one experiment in which we train a Mujoco’s Hopper agent on our RL forward model, performing the tests of this training in the real Mujoco’s environment. The main idea behind this is to verify the results of our research testing the trained forward models (both SL and RL) in real envs.  In this sense, we extended the conclusions with the results.
>
> >It's unclear how the authors proposed method would perform (RL calculated forward models) would perform in relevant application settings, such as model-based RL. The authors allude to this point in the conclusion and future work, but the paper would be greatly strengthened if indications of performance would be included.
>
> * Answer: Regarding the applications, we mention that these forward models could help us to have a simulated environment (Gym I.e.) trained in a batch/offline way robust enough to be the base of an RL standard training setup. This is especially important in environments that cannot be used as a training environment, such as some industrial facilities.
>
> >Following on the above two points, the significance of the overall results is unclear. It would be good to add further clarity to this, ideally by additional experiments in an application setting or an indication of the advantages this method could provide. :
>
> * Answer: As we have mentioned, having a good forward model to be used as a training environment is one of the main goals of this research. Our approach has been demonstrated as a robust starting point, a thing that is noticeable in the experiments where we compare SL and RL approaches in the same well known Mujoco envs. It is remarkable that RL performs better than SL in some circumstances such as longer trajectories. The next goal (future work) will be to train agents or our forward model in order to verify the robustness of these models.
>
> With regards to your questions:
>
> >What is dt in Table 1? I do not see it defined in the text near it. :
> *  dt is an internal fixed metric of Mujoco’s environment (integration step), that is different for each one of them. It determines the amount of time of every timestep, and it is necessary to derive velocities from positions and vice versa.
>
> >Where is the reward in Figure 2?
> *  Is it the same as the RMSE, which would be confusing since it should be the signal loss, correct? : The reward is according to Eq. 15 and Eq. 16. It is basically a signal similarity function at end of rollouts (based on three terms), and RMSE otherwise.
>
> >Could you clarify test rollout setting -  is the model just predicting further into the future?:
> * ***YES*** at every time step, we extract an initial state and a set of actions performed in the original datasets. From this initial state and performing all the actions, the forward model predicts the variation of the variables in order to make the agent evolve to the following state. The forward model compounds every prediction into the predicted trajectory until the rollout max length is achieved.
>
> >It's unclear what Figure 4 and Figure 5 are trying to show. Could you provide further description?
> * Both figures represent one Hopper random episode in which all the position variables (a total of 6) have been predicted according to the dataset actions using forward models (SL and RL approaches).

---

### Official Review · Reviewer_xo1S · 2022-10-23

**Confidence:** 5
**Correctness:** 3
**Technical Novelty And Significance:** 2
**Empirical Novelty And Significance:** 2
**Recommendation:** 3

**Clarity, Quality, Novelty And Reproducibility:**

The clarity is generally acceptable, albeit there is an overly aggressive use of bullet points which makes at times hard to follow the discussion.

The quality of the work could be improved. since there is no comparison with existing relevant work on the topic.

I am not aware of any work explicitly using reinforcement learning for learning models of the dynamics.

There seem to be no clear reproducibility issues.

**Strength And Weaknesses:**

Strength:
- The problem of learning multi-step models is important, especially compared to the amount of work that explicitly addresses it;
- The motivation section, while being a bit overly schematic, explicitly states the goal of the work;
- The approach seem to have an advantage compared to supervised learning, in the limited evaluation it was subjected to.

Weaknesses:
- The relationship with existing work is not clearly presented. Essentially presenting a method for learning a multi-step model, the paper should position itself, and compare experimentally, to other work in this space. A few works that come to my mind are the investigation of Erin Talvitie (e.g., https://arxiv.org/abs/1612.06018) and Kavosh Asadi (e.g., https://arxiv.org/abs/1905.13320) and their collaborators, as well as work more based on the option perspective (e.g., https://arxiv.org/abs/2108.03213). Without a comparison with at least some of these works, it is not easy to interpret the actual contribution of the paper;
- In particular, there is a tight relationship between teacher/student forcing in RNN training and multi-step models. To put things into perspective, using reinforcement learning to train a model of the dynamics resembles using reinforcement learning for training RNNs. In the lack of a direct comparison or a justification for this, one might wonder if this is an unnecessary overhead over more traditional methods for learning multi-step models;
- More broadly, showing that one can obtain more accurate models without showing how useful are they for reinforcement learning can generally be misleading: as it has been shown several times in recent work, an accurate model of the dynamics does not necessarily mean better resulting policies after policy optimization (see workshop on the topic: https://icml.cc/virtual/2022/workshop/13463).

**Summary Of The Paper:**

The paper presents an approach based on using reinforcement learning for learning dynamics models in simulated robotics benchmark. A method based on SAC for learning multi-step models is implemented and compared to traditional supervised learning in terms of long-term accuracy in the predictions.

**Summary Of The Review:**

Overall, the approach is a potentially interesting direction for effective multi-step model learning. However, there is a lack of comparison with related algorithms, both in the text and in the experiments, as well as an insufficient investigation of the implication for actual policy optimization. Thus, I for now recommend rejection.

---

> ### Author Response · Authors · 2022-11-18
> **Thanks for your review and time spent. a new version is coming hoping this clarify the reading and appreciation of the paper**
>
> First, we are very grateful for your review and appreciate the time and effort put into it. As for your comments regarding potential areas of improvement:
> Weaknesses:
>
> >The relationship with existing work is not clearly presented. Essentially presenting a method for learning a multi-step model, the paper should position itself, and compare experimentally, to other work in this space. A few works that come to my mind are the investigation of Erin Talvitie (e.g., https://arxiv.org/abs/1612.06018) and Kavosh Asadi (e.g., https://arxiv.org/abs/1905.13320) and their collaborators, as well as work more based on the option perspective (e.g., https://arxiv.org/abs/2108.03213). Without a comparison with at least some of these works, it is not easy to interpret the actual contribution of the paper;
>
> * Answer: Thanks for your comments. In this new version, we have reformulated and expanded the introduction and motivation sections (Section 1 and Section 2) with the aim of improving our research’s context and making things clearer. This includes the addition of new references similar to the works you refer to and to others that we think are better fitted for this paper.
>
> >In particular, there is a tight relationship between teacher/student forcing in RNN training and multi-step models. To put things into perspective, using reinforcement learning to train a model of the dynamics resembles using reinforcement learning for training RNNs. In the lack of a direct comparison or a justification for this, one might wonder if this is an unnecessary overhead over more traditional methods for learning multi-step models;
>
> * Answer: Please, refer to the previous answer of this response.
>
> >More broadly, showing that one can obtain more accurate models without showing how useful are they for reinforcement learning can generally be misleading: as it has been shown several times in recent work, an accurate model of the dynamics does not necessarily mean better resulting policies after policy optimization (see workshop on the topic: https://icml.cc/virtual/2022/workshop/13463).
>
> * Answer: We totally agree with your point of view. Now in the updated version, you can find this evaluation in Section 4.2 on page 8. In short, we first trained RL agents with the RL forward models (and the SL forward models as the baseline) and tested them on the real MuJoCo environment. We obtained promising results compared to the SL baseline.
>
> Clarity, Quality, Novelty And Reproducibility:
>
> >The clarity is generally acceptable, albeit there is an overly aggressive use of bullet points which makes at times hard to follow the discussion.
>
> * Answer: Thanks for the suggestion. In the new updated version, we now have reformatted all bullet points into text and made them easier to read and follow.
>
> >The quality of the work could be improved. since there is no comparison with existing relevant work on the topic.
>
> * Answer: Yes of course and thanks for helping us in this. we reworked a lot the writing style hopping this help in understanding the paper and we did a more thorough review following the tree of publications to include the relevant references for giving the wright context.
>
> Thanks you so much
> Kind regars,
> The authors.

---

### Official Review · Reviewer_C9nJ · 2022-11-03

**Confidence:** 4
**Correctness:** 2
**Technical Novelty And Significance:** 2
**Empirical Novelty And Significance:** 2
**Recommendation:** 1

**Clarity, Quality, Novelty And Reproducibility:**

The paper is hard to follow and the author's meaning often unclear. It seems like many terms are used very loosely vaguely or even incorrectly (e.g., bootstrapping). At times, it feels like the authors have something specific in mind but use general terms to reference them, making it difficult to understand the argument being discussed (e.g., "bootstrapped rollouts" in the context of learning forward models). Overall, the authors would benefit from making sure to rigorously define terms before using them when motivating and explaining the work.

As for novelty, I'm not aware of any work proposing to learn models by formulating an MDP in this way. Some concepts of duality of automata theory [1], MDPs and POMDPs [2] might be of interest to the authors since the consider transformations with a similar flavor. Although not immediately relevant to this work as it is, it might inspire future work on this idea of formulating model-learning as a different RL problem.

The authors should consider looking into [3] and citing what is relevant since it covers some impactful work on learning models, by say, solving fixed-points [4].

[1] Brzozowski JA. Canonical regular expressions and minimal state graphs for definite events. InProc. Symposium of Mathematical Theory of Automata 1962 (pp. 529-561).
[2] Hundt C, Panangaden P, Pineau J, Precup D. Representing Systems with Hidden State. InAAAI 2006 Jan 1 (pp. 368-374).
[3] Rust J. Structural estimation of Markov decision processes. Handbook of econometrics. 1994 Jan 1;4:3081-143.
[4] John Rust. Maximum likelihood estimation of discrete control processes. SIAM journal on control and optimization, 26(5):1006–1024, 1988.

Minor comments
============

Eqs. (1-4) are very confusing. The order of operations/relations seems ambiguous. Are those even equations?

Eq. (11), how is $Z_E$ an open choice if it respects equality with equation above it?

p. 5, " A specific Gym environment MF has been developed", what does that mean?

p. 5, showing what the update rules of SAC on $M_F$ in terms of states/actions of $M$ would be helpful.

Eq. (15), what is the motivation for this? What is the derivative taken with respect to?

p. 6, last para, what is the "true dataset"?

p. 7, how is the training data generated?

Figure 4 and 5, what the rollout step value (50 and 500) represent here? Why does the shown trajectory both seem of the same length?



**Strength And Weaknesses:**

The basic idea is original and could be interesting. However, there are are several limitations in it's current iteration.

It's not quite clear what inherent advantage there is to reformulate the model-learning problem as an RL problem. What property of this newly defined RL problem makes it better behaved than an equivalently expressive supervised learning approach? The authors seem to imply that one-step prediction errors are all that supervised learning could on this task, but there are a plethora of ways to formulate this learning problem. I don't see what would be the fundamental difference between "rollout learning" and a supervised learning approach which optimizes error defined on entire trajectories. The authors should take more time expanding their arguments and accompany them with explicitly written out equations.

One major limitation of the proposed method is that it doesn't address the inherent difficulties of learning stochastic transitions and instead limits itself to additive gaussian transitions. Could the main idea not be applied to discrete MDPs? Exploring the subject in illustrative discrete MDPs would help provide insight while also showing that the method depend on having limited stochastic transition models.

Similarly, the assumption that states support addition or subtraction greatly limits it's applicability to domains that have discrete or hybrid state-space.

Minor comments
=============

I could not find the tips cited as Fleming 2018.


**Summary Of The Paper:**

The authors propose to learn forward models using RL methods on a new MDP who's states correspond to state-action tuples and who's actions correspond to additive changes in state. From this idea, they propose to use soft actor-critic to learn a gaussian transition model using negative prediction errors as reward with an additional reward injected at the end of rollouts related to the some negative error defined on the whole trajectory. This approach is then applied to 3 MuJoCo domains and compared to a supervised learning baseline.

**Summary Of The Review:**

The authors proposes an original idea but the clarity of the paper prevents me from appreciating its possible benefits.

---

> ### Author Response · Authors · 2022-11-18
> **We are doing our best to improve it! thanks a lot!**
>
> Dear reviewer, thank you very much for your kind comments and the time and effort spent in reviewing our paper.
>
> *Weaknesses*
>
> > It's not quite clear what inherent advantage there is to reformulate the model-learning problem as an RL problem. What property of this newly defined RL problem makes it better behaved than an equivalently expressive supervised learning approach? ...
>
> * Answer: Regarding the advantage of the RL approach versus an SL approach, we would like to first point out that our objective has always been to explore a different paradigm for a problem which in our view naturally fits under the RL framework, especially when dealing with multistep and sequential trajectories.  We are completely aware of the potential and extensive development of SL techniques on such type of problems, which we also have tested. In this new version, we have reformulated and expanded the introduction and motivation sections (Section 1 and Section 2) with the aim of improving our research context and making clearer the motivation for this line of research.
>
> >One major limitation of the proposed method is that it doesn't address the inherent difficulties of learning stochastic transitions and instead limits itself to additive gaussian transitions. Could the main idea not be applied to discrete MDPs?...
>
> * Answer: We appreciate your comment, and we totally agree that there are still room to expand on this line of research, one of them would be to extend it to other types of domains both in states and actions (discrete, hybrid, large, non-additive, delayed effects, etc) but we put the focus on the type of problem which fits both on the well-know mujoco suite control problems and our own type of problem that motivates our research on real industrial control problems. Nevertheless, we will consider your comments to continue our research along these lines.
>
> *Novelty*:  We appreciate the references provided by the reviewer which for sure will be very helpful to us to expand our knowledge on this topic and inspire us on this line of research that we are willing to continue.
>
> *Minor comments*:
>
> >I could not find the tips cited as Fleming 2018.
> * Derivation and method section (3) has been rewritten to accommodate some of the reviewers’ comments and concerns. Reference to this citation was removed.
>
> >Eqs. (1-4) are very confusing. The order of operations/relations seems ambiguous. Are those even equations?
> * Answer: Eqs (1-4) refer to the translation of an original MDP problem (eq 1) into a translated MDP problem (eq 2) that we have made in order to formulate the forward learning problem as a control RL problem, that is, a problem which objective is to find optimal policy (eq 3) that is close as possible to optimal policy (eq 4). For clarity purposes we have just rewritten them and keep only two expressions.
>
> >Eq. (11), how is an open choice if it respects equality with equation above it?
> * Answer: Eq. (12) defines the rollout loss between two trajectories in a more general way than Eq(11) that refers to the more common way to define it based on RSME. So both of them are possible, being Eq (12) an open choice if one would like to consider another type of signal divergence measure as it has been done in the experimental part.  We have included an ***or*** among both equations to point out that any of the choices are possible.
>
> >p. 5, " A specific Gym environment MF has been developed", what does that mean?
> * Answer: It refers to the gym environment that codified the MDP for the forward learning problem on which a RL agent is trained
>
> >p. 5, showing what the update rules of SAC on in terms of states/actions of would be helpful
> * Answer: Although we understand that it would be beneficial to be able to explain more about the selected algorithm, it was not possible due to article length restrictions. Nevertheless, we have included the reference.
>
> >Eq. (15), what is the motivation for this? What is the derivative taken with respect to?
> * Answer: Please refer to the updated version (page 3) where we have improved explanation and motivation for this measure.
>
> >p. 6, last para, what is the "true dataset"?
> * Answer: It refers to the DRL4 datasets. It has been rewritten more precisely in order to avoid confusions in the paper
>
> >p. 7, how is the training data generated?
> * Answer: Explanation can be found in Page 4, Section 4 Experimental evaluation, first paragraph
>
> >Figure 4 and 5, what the rollout step value (50 and 500) represent here? Why does the shown trajectory both seem of the same length?
> *Answer: Figure 4 has been updated with more experimentation on different rollout lengths as well as its explanation. Rollout lengths refer to the length of the trajectory generated sequentially by means of the RL agent.

---

### Official Review · Reviewer_wuwo · 2022-11-04

**Confidence:** 4
**Correctness:** 3
**Technical Novelty And Significance:** 3
**Empirical Novelty And Significance:** 3
**Recommendation:** 6

**Clarity, Quality, Novelty And Reproducibility:**

The logic is clear, but the presentation is not well-polished. The ideas are novel. There seems no big issue with reproducibility.

**Strength And Weaknesses:**

Strengths:
1. The research problem is important and the recursive reformulation of the forward model as the summation of the previous state and a derivate approximation is novel. System identification is one of the most important topics in the modern automatic control area. Vanilla supervised methods are intuitive and straightforward, yet suffer from several challenges, e.g., rollout testing and loss accumulation. The proposed forward model has advantages in solving the aforementioned challenges.
2. The justification of the motivation to learn forward models with RL and the design of function approximation is reasonable.
3. The experiments are supportive.

Weaknesses:
1. The major difference is to reformulate the system identification problem into a recursive relation, and take accumulation error into account. In my opinion, the proposed method still uses a supervised learning strategy. Reinforcement learning here only plays the role of an optimization tool, which optimizes the supervised loss.
2. The paper is not well-polished. Subfigures in Figure 3 is massive and some figures overlap with each other.

**Summary Of The Paper:**

This paper proposes to regard the system identification problem as a reinforcement learning problem. Compared with traditional supervised solutions, the proposed reinforcement method has advantages over delayed effects, high non-linearity, non-stationarity, partial observability, and error accumulation when using bootstrapped predictions.

**Summary Of The Review:**

This paper proposed a wrapper framework to solve system identification problems as reinforcement learning tasks. The idea of recursive relation between states is beneficial to dealing with accumulative errors in rollouts. The logic is clear, but the presentation needs polish.

---

### Decision · Program_Chairs · 2023-01-20

**Decision:**

Reject

**Justification For Why Not Higher Score:**

NA

**Justification For Why Not Lower Score:**

NA

**Metareview: Summary, Strengths And Weaknesses:**

This paper proposed to formulate and solve system Identification as an RL problem. The idea is interesting but the paper is premature for publication. In particular, several drawbacks are raised by reviewers, e.g. insufficient discussion of related works, poor writing, and weak experiments. Therefore, I vote for rejection.